# A Homeostatic Model of Subjective Cognitive Decline

**DOI:** 10.3390/brainsci8120228

**Published:** 2018-12-19

**Authors:** Akiko Mizuno, Maria Ly, Howard J. Aizenstein

**Affiliations:** 1Department of Psychiatry, University of Pittsburgh, Pittsburgh, PA 15213, USA; Ly.Maria@medstudent.pitt.edu (M.L.); aizen@pitt.edu (H.J.A.); 2Department of Neuroscience, University of Pittsburgh, Pittsburgh, PA 15213, USA; 3Department of Bioengineering, University of Pittsburgh, Pittsburgh, PA 15213, USA

**Keywords:** subjective cognitive decline, preclinical dementia, fMRI, compensation

## Abstract

Subjective Cognitive Decline (SCD) is possibly one of the earliest detectable signs of dementia, but we do not know which mental processes lead to elevated concern. In this narrative review, we will summarize the previous literature on the biomarkers and functional neuroanatomy of SCD. In order to extend upon the prevailing theory of SCD, compensatory hyperactivation, we will introduce a new model: the breakdown of homeostasis in the prediction error minimization system. A cognitive prediction error is a discrepancy between an implicit cognitive prediction and the corresponding outcome. Experiencing frequent prediction errors may be a primary source of elevated subjective concern. Our homeostasis breakdown model provides an explanation for the progression from both normal cognition to SCD and from SCD to advanced dementia stages.

## 1. Introduction

Subjective Cognitive Decline (SCD) refers to an individual’s perception that their cognitive performance has declined, despite having no significant objective cognitive impairment. SCD may reflect one of the earliest signs of dementia, as it is a risk factor for developing mild cognitive impairment (MCI) and Alzheimer’s disease (AD) [1,2]. However, SCD is quite understudied—which mental processes lead to SCD and the neural basis of SCD are yet to be understood. Here, we will provide a narrative review of the current literature for biomarkers, and the functional neuroanatomy associated with SCD. Then, we will propose a new model that integrates existing findings in SCD into a new neural system dysfunction model, which involves heightened prediction of error-signaling and homeostatic breakdown.

## 2. AD Biomarkers (Neurodegenerative Factors) and SCD

In order to investigate whether SCD represents a pre-clinical state of AD, the relationships between AD biomarkers in individuals with SCD were examined. In the traditional AD biomarker cascade, amyloid (Aβ) accumulation occurs prior to neurodegeneration and cognitive decline [3]. The most promising evidence that SCD precedes MCI and AD is that Aβ deposition is associated with SCD symptom severity, but not with objective memory performance [4,5,6]. Vogel and colleagues [7] found that amyloid status predicted future cognitive decline (on average, four years) among individuals with SCD. However, a larger longitudinal study [8] concluded that the Aβ-status by itself does not predict the progression of AD over a relatively brief time span of 2.5 years. More longitudinal studies are necessary to better understand the relationship between Aβ and objective cognitive decline in SCD. In addition to the accumulation of amyloid plaques, AD is also associated with neurofibrillary tangles composed of tau protein. Tau accumulation is believed to be more closely related to neurodegeneration and cognitive decline in AD, as compared with amyloid [9]. Tau has been shown to be more associated with SCD than with non-amnestic MCI [10]. Thus, like Aβ, the tau markers support the role SCD as an AD risk group.

Brain atrophy and white matter hyperintensities have been also reported in SCD, as seen in the early pre-clinical stages of AD progression. Several studies consistently reported cortical volume loss and thinning in the medial temporal regions in those with SCD [11,12,13,14,15], indicating the decreased structural integrity of the memory system. Longitudinal observations also reported the association between atrophy and future cognitive decline in SCD [16,17]. Whole-brain analysis by Verfaillie and colleagues [17] suggested that a steeper decline in cognition was not only associated with a thinner cortex of the temporal region but also the frontal and occipital cortices in SCD. Increased amounts of white matter hyperintensities in widespread regions have also been reported in SCD [18,19]. These studies provide support for the idea that SCD may be an early transitional stage prior to the onset of dementia (i.e., MCI and AD), especially as seen with the perturbations in the memory systems.

## 3. Functional Neuroanatomy and Compensation Theory in SCD

The neural basis of elevated subjective concern for cognitive decline among older individuals with normal cognition is the least investigated, but it represents a growing area of research (Table 1). Four functional magnetic resonance imaging (fMRI) studies investigated brain activation during memory-related tasks [20,21,22,23]. Most of these studies did not find group differences between participants with and without SCD in the behavioral performance of the task, but observed different functional brain activation patterns. The first study by Rodda and colleagues [23] measured brain activity while participants were encoding a list of semantically related words, which were later tested through a recognition paradigm. Whole-brain analysis demonstrated increased activation in the lateral part of the prefrontal cortex (PFC) in those with SCD. The level of the PFC activation was positively correlated with task performance. The authors interpreted that increased PFC activation served as neural compensation for the decreasing function of the primary hippocampal memory system, as indicated in previous structural imaging studies (summarized in the previous section). To test this compensation hypothesis, Erk et al. (2011) [20] investigated activation in the hippocampus and the PFC during memory encoding of faces and associated occupations, through a region-of-interest approach. Their results demonstrated decreased activation in the hippocampus, and increased activation in the dorsolateral PFC (dlPFC). Task performance was positively correlated with dlPFC activation only in the SCD group, which provided support for the compensation hypothesis in SCD.

Hu et al. (2017) [22] utilized a task that emulated memory processes that were relevant to activities in daily life (future-oriented choice tasks). Brain activation was measured, while participants were required to select an immediate or delayed reward regarding a personally relevant episodic future event. Successful selection of the future-oriented choice (i.e., delayed reward) over the immediate reward requires the crucial involvement of episodic memory and valuation systems [26]. Unlike other studies, their study is the only one which observed group differences in task performance. The SCD group showed reduced preference for future-oriented choice, which was previously demonstrated in those with MCI [27]. A priori region-of-interest analysis showed that only participants in the control group showed an association between greater hippocampal activation and more future-oriented choices. The whole-brain analyses found reduced activation in medial frontal regions (medial frontal pole and anterior cingulate cortex (ACC)) and the insula in the SCD group, suggesting diminished valuation functioning. The authors suggested that reduced involvement in the episodic memory and valuation system in SCD during the decision-making process may reflect the attenuating attention and subjective evaluation system.

To address the possibility that increased PFC activation reflects a general cognitive processes rather than only memory encoding, Hayes et al. (2017) [21] used an event-related design to compare brain activation for high-confidence successful recall versus failed recall. The SCD group showed an increased activation for failed recall (i.e., negative subsequent memory) in the posterior areas (occipital, superior parietal, and precuneus). They also regressed activation on a continuous SCD symptom severity to identify the neural correlates of SCD, by combining participants in both groups. Participants with more severe SCD symptoms showed increased activation for failed recall in both the frontal and posterior nodes of the default mode network, which normally suppress its activity during cognitive tasks. They concluded that individuals with SCD rely on the altered neural system for successful memory encoding to maintain normal cognitive function.

Cognitive concerns in SCD mainly reflect the perceived decline of memory function, but the other domains of cognitive function may also contribute to elevated concerns [28]. There are two fMRI studies with non-episodic memory tasks. Dumas and colleagues [25] measured brain activation in SCD by using the n-back working memory task. Although this study was limited to females (i.e., a comparison between those with and without cognitive concerns among postmenopausal women without hormone therapy), their study is the first to report that activation increased with increasing cognitive load/effort among those with SCD. These effects were found in the extended working memory system, including middle frontal (BA 9/10), anterior cingulate cortex (BA 24/32), and the precuneus (BA 13). Both groups showed the same levels of behavioral performance. Another study by Rodda et al. (2011) [24] investigated brain activation during a divided attention task, where participants with SCD were required to respond to target stimuli while processing sequences of both visual and auditory information. Behavioral performance did not show group differences, but the SCD group demonstrated increased activation in two medial posterior regions: one in the cerebellum, and another in the thalamus, extending to the posterior cingulate cortex and the medial temporal lobe (hippocampus and parahippocampus). These two studies are consistent with the idea that early functional changes (i.e., increased activation) in executive function may manifest in SCD, despite the lack of impairment in behavioral performance or in the neuropsychological tests.

In summary, previous fMRI studies, along with the structural imaging studies (summarized in the previous section) have suggested three neural phenomena in SCD: (1) loss of integration of the memory system, (2) compensatory hyperactivation in the prefrontal cortex or the use of other alternative neural resources to maintain normal performance, and (3) decreased prefrontal activation for subtle yet declining higher-order cognitive functions. In other words, the direction of neural activation (increased or decreased) observed in SCD depends on whether the expected level of performance can be maintained. However, it is important to note that the sample sizes and statistical power in these studies were relatively low; likewise, inconsistencies between studies may have been partially due to sampling error. To extend our understanding of the functional neuroanatomy of SCD, more mechanistic and cohesive frameworks that can provide explanations for dysfunctional processes are necessary, regardless the level of task performance or cognitive domain. Furthermore, it is not yet understood which specific cognitive processes rely upon compensation, and how these processes are directly associated with SCD symptoms.

## 4. Current Theories of the Neural Basis of SCD

Compensatory hyperactivation of the prefrontal region is currently the most popular theory of the neural basis of SCD. Another theory is brain reserve [29], which proposes a structural basis for functional compensatory capacity. Alternatively, the dedifferentiation [30], a loss of specialization of neural function resulting in diffuse brain activations, is a theory which may explain hyperactivation in the prefrontal cortex. All of these versions of compensatory hyperactivation describe only the temporal transition from the pre-SCD state to the SCD state, and do not describe the dynamics of post-SCD neurodegeneration. Further, they do not describe how hyperactivation may contribute to post-SCD decline via harmful biological effects on the neural system, such as neurotoxicity or excitotoxicity. Here, we introduce homeostasis breakdown, a new mechanistic framework for comprehensive temporal dynamics in SCD and progression to AD.

## 5. Background of the Prediction Error Theory

Our brain functions as a statistical optimization engine that constantly makes implicit predictions of sensory inputs [31,32]. That is, rather than passively receiving sensory information, it is actively making inferences. These inferences are propagated as predicted expectations to heteromodal association areas and the PFC. These expectations are compared with the current environment, and a behavior is chosen. Moreover, the difference between predicted and observed behavior is used as learning signal to adapt for better performance in the future. This constant process of comparing internally generated predictions with external reality is called “predictive coding” [33]. This is why we think of the brain as learning and adapting across all behaviors.

Prediction errors refer to the mismatch between the internally generated prediction and the external reality. The most prominent brain region that responds to such errors is the dorsal anterior cingulate cortex (dACC). Both animal and human studies have demonstrated the increased activity in dACC to response to prediction errors [34,35]. Similar terms for prediction errors are conflict [34,36] and free-energy [37]. Although there is a general consensus that dACC mediates error-related signals; neuroscientists have different opinions about the specific processes in the dACC and the primary goal of the function [38]. Nonetheless (regardless of the diverse terminology and theories of dACC function), the minimization of prediction error is a core organizing principal for computational function at the local neural circuit. This error minimization optimizes our internal predictions, which facilitates successful goal-directed behaviors and survival.

## 6. Prediction Error and SCD Symptoms

Suppose a man who is very experienced with a computer notices that he is making more typing errors. If he experiences subtle yet frequent errors between his prediction (“I thought I typed out ‘experience’”) and actual outcome (“I accidentally typed ‘exprience’”), his level of SCD symptoms may rise. This type of error signal raises the activation in the dACC, resulting in varying levels of awareness. We believe that an uncharacteristic accumulation of implicit errors may gradually lead to more effort being required to maintain cognitive performance, and then to having more explicit levels of error awareness. Another example that may highlight the relationship between error monitoring and the experience of cognitive tasks being more effortful in SCD would be the following: an individual may be accustomed to finishing the “New York Times” crossword puzzle in 20 minutes (their prediction). However, if that individual finds that they are now needing 30 minutes or more to complete the puzzle, this reflects both a prediction error and the experience of increased effort. Such awareness of errors could occur not only in memory, but rather across multiple cognitive domains, including attention, task switching, language, and mathematical operations.

Awareness of one’s internal cognitive system is called metacognition. According to Nelson (1990) [39], metacognition has two primary operations: monitoring and control. Monitoring refers to the introspection of incoming sensory information and one’s own performance, whereas control refers to an allocation of an action (i.e., self-regulation). These two operations are independent, but reciprocally interacted. Both the prediction error [31,32] and conflict monitoring [34,36] frameworks provide mechanistic models for the monitoring of errors or conflicts. These models, however, differ in the level of information processing that they are meant to explain. Prediction error refers to the process that can occur throughout the cortical network. Prediction errors provide a signal that our internal model need to be updated, and the signal is generated by distributed processes of our incoming sensory information [40]. These local prediction errors influence the local Hebbian learning model [41]. In this way, correct predictions are strengthened, and incorrect predictions are weakened [42]. On the other hand, conflict-monitoring framework refers to how the more extended controlled yet implicit cognitive process integrates generated error signals, such as prepotent response suppression. Unlike prediction error, which is a general network learning signal, conflict monitoring refers to the specific monitoring and control functions in the dACC. The prediction error [31,32] process may specifically infer to the earlier operation of monitoring, whereas conflict-monitoring [34,36] may associate with both monitoring and control operations, suggested in Nelson’s framework [39].

In the framework of metacognition, SCD can be interpreted as the impairment of both monitoring and the early stages of control. The accumulated subjective experience of prediction errors and perceived increase in an effort to complete tasks may eventually lead to the elevated self-awareness of cognitive decline, as reflected in SCD. As these individuals develop frank cognitive decline, conflict-monitoring processes may then decrease, leading to an opposite pattern of decreased self-awareness of cognitive functioning. Experimental tasks that are sensitive to the suppression of the inappropriate responses to implicit prediction error may be able to capture an earlier objective process of decline in SCD. Furthermore, neuroimaging studies with these tasks will provide detailed neural mechanisms of dysfunctional metacognition in SCD.

## 7. Prediction Error and SCD Characteristics

Previous studies suggest that the cognitive implications of SCD symptoms depend on the level of education achievement [28]. Since higher level of education is considered to be a marker of cognitive reserve, Stern [43,44] postulated that cognitive reserve provides resilience to neurodegeneration. Levels of cognitive reserve may represent the sensitivity to prediction errors, and the utility of the error signals. Individuals with high cognitive reserve may be highly sensitive to prediction errors, and interpret them as important learning signals to update the internal model. These individuals constitute a lifestyle of using these learning signals more frequently and effectively to make higher achievements, resulting in having a high cognitive reserve.

Individuals with SCD are often highly anxious, and characterized by their tendency to worry [1], usually expressed as high neuroticism in the Big Five personality trait model [45]. It has been demonstrated that individuals with high neuroticism are highly sensitive to prediction errors [46]. In the course of progression of neurodegeneration, these individuals may start noticing prediction errors earlier than those with low neuroticism. The frequent experience of errors may not only raise an awareness, but may also elicit concern. Individual with high neuroticism may then interpret the perceived errors as important learning signals, resulting in symptoms of SCD.

The high prevalence of depressive symptoms is another characteristic that has been reported consistently in SCD [1]. Neuroticism is also highly associated with depression; however, the symptoms associated with neuroticism and depression may relate to different aspects of SCD. Neuroticism may serve as a predictor of how an individual may interpret prediction errors, whereas depressive symptoms may reflect the affective response to an individual’s interpretation of their prediction errors. Depressive symptoms in SCD, therefore, may be translated as a negative affective response (i.e., sad feeling) to frequently experiencing errors (i.e., the “monitoring” component of metacognition), leading to the persistence of depressive moods over time [47]. Depressive symptoms may also be a form of adjustment disorder, where an individual may have an emotional reaction to their new experience of difficulty in both internal prediction and performance (i.e., the “control” component of metacognition).

## 8. Homeostasis Breakdown

Homeostasis—or homeostatic regulation—is the ability to maintain stability and equilibrium of the system. As a classical example, the stability of our body temperature is a consequence of homeostatic processes that coordinate the activity of muscles, blood vessels, and sweat glands. When a cold environment decreases body temperature, the hypothalamus releases a signal to the skeletal muscles, promoting shivering as a mechanism of thermogenesis and a signal to the blood vessels to increase resistance of blood flow (i.e., vasoconstriction). Both of these responses minimize heat loss, helping to maintain body temperature.

Prior work by Li et al. has suggested that homeostatic dysregulation in multiple systems occur in aging, and may also serve as a key contributor to the biological mechanisms of aging [48]. While the authors have provided evidence in the systems of lipids, immune function, oxygen transport, liver functioning, vitamin levels, and electrolyte levels, they suggest that homeostatic dysregulation is not limited to these systems, and may occur in other systems in aging.

Thus, we propose that compensatory hyperactivation may represent a homeostatic process that serves to maintain the stability of cognition in a changing neurobiological environment. Homeostasis in the context of cognition serves to maintain cognitive functioning, despite the presence of neurodegeneration. Neurodegeneration may lead to prediction errors and corresponding SCD symptoms, much like the body temperature falling just enough to cause a sensation of coldness. Finally, the compensatory hyperactivation is one of the main homeostatic processes that we are currently aware of in cognition, and this is analogous to the onset of vasoconstriction of blood vessels to prevent hypothermia.

However, homeostatic processes can have negative side effects. For example, extreme vasoconstriction for an extended period of time can lead to vascular cell loss. Similarly, compensatory hyperactivation may lead to glutaminergic excitotoxicity, which may lead to neuronal death [49] or the production of Aβ [50]. Thus, although homeostasis can slow the onset of cognitive decline, this may come at the cost of negative side effects that weaken the core cognitive infrastructure. This may explain why individuals with SCD tend to experience a relatively rapid decline into AD [51].

This homeostatic model of SCD represents an extension of the prevailing compensatory hyperactivation model, as it may provide a mechanistic explanation for differing levels of neural activity. The compensation model is unclear as to what entity “drives” the change in activation levels, whereas the homeostatic model provides a dynamic control system that maintains cognitive functioning that is dependent on prediction error and conflict monitoring.

## 9. Future Directions

Clinicians do not yet have a standard intervention protocol for individuals with SCD. If the neural basis for SCD were better understood, an effective intervention may be developed. A recent meta-analysis of experimental interventions for SCD suggested that cognitive restructuring therapies may improve metacognition (i.e., alleviate self-perceived cognitive challenges) [52], indicating that SCD could be a modifiable risk factor of dementia. Alleviating SCD symptoms, along with associated psychological distress may slow neurodegeneration, such as atrophy and Aβ accumulation, by reducing hyperactivation.

More studies investigating the markers of neurotoxicity in SCD are necessary to provide basic evidence for psychotherapeutic interventions in the earliest stages of dementia. In this narrative review, we have proposed a homeostatic prediction error model for understanding how the progression of neural system dysfunction can manifest as SCD. fMRI studies can help in validating that increases in error prediction, and conflict monitoring is central to the subjective perception of cognitive impairment. Further research is also necessary to identify the working elements (neural control system, sensors, set point, etc.) of the homeostatic prediction error model. Moreover, fMRI studies coupled with behavioral interventions can be used as an outcome measure. This would help in the development and adaptation of behavioral interventions targeting SCD, and potentially mitigating the accelerated neurodegeneration that may be associated with hyperactivated stressed system that is out of homeostatic balance.

## 10. Conclusions

In common scientific practice, the term *subjective* may generally be disfavored, because it connotes a lack of objectivity, as the self-assessments that are used to diagnose SCD, presumably include individual biases. However, the presence of SCD symptoms appears to contain valuable information regarding cognitive decline over time and underlying neurophysiological pathologies. More studies and theoretical frameworks that can comprehensively explain temporal dynamics, including the positive and negative by-products of compensatory hyperactivation, are necessary. In this review, we proposed that prediction error, a metacognitive process, potentially leads to SCD symptoms. We also introduced homeostatic breakdown as a new framework that incorporates and integrates the current findings with the new prediction error perspective, to describe the cascading effect of neurodegeneration and cognitive decline in SCD. This framework has the potential to motivate new standard therapies for SCD that focus on alleviating not only the subjective symptoms, but also slow the progression of dementia, due to neurotoxicity from compensatory hyperactivation.

## Figures and Tables

**Table 1 brainsci-08-00228-t001:** Summary of fMRI studies in SCD. Most of these studies demonstrate a pattern of regional hypoactivation, associated with hyperactivation elsewhere. This hyperactivation has been interpreted by authors as a compensatory response to the hypoactivation. Abbreviations: SCD: subjective cognitive decline, PFC: prefrontal cortex, BA: Brodmann area, DLPFC: dorsolateral prefrontal cortex, ROI: region-of-interest, SPL: superior parietal lobe, PCC: posterior cingulate cortex, DMN: default mode network, VMPFC: ventromedial prefrontal cortex, ACC: anterior cingulate cortex, MFG: middle frontal gyrus, WM: working memory.

Reference	fMRI Task	Participants	Hyperactivation (SCD > Control) or Positive Correlation with SCD Symptoms	Hypoactivation (Control > SCD) or Negative Correlation with SCD Symptoms	Behavioral Performance
Rodda et al. (2009) [23]	Memory encoding	10 memory clinic SCD vs. 10 controls (age: 64.2 vs. 68.0)	L PFC (BA6/9/44/46)		(1) No group differences in behavioral performance (2) Positive correlation between PFC activation and recognition performance in both groups.
Erk et al. (2011) [20]	Memory (encoding, recall, recognition) and working memory (n-back)	19 memory clinic SCD vs. 20 controls (age: 68.4 vs. 66.8)	R DLPFC during recall (ROI analysis)	Hippocampus during recall (ROI analysis)	(1) No group differences in behavioral performance (2) Positive correlation between DLPFC activation and recognition performance in SCD. (3) Positive correlation between hippocampal activation and recognition performance in controls
Rodda et al. (2011) [24]	Divided attention	11 memory clinic SCD vs. 10 controls (age: 64.6 vs. 68.0)	L medial temporal, bilateral thalamus, PCC, caudate		(1) No group differences in behavioral performance
Dumas et al. (2013) [25]	Working memory (n-back)	Postmenopausal women: 12 cognitive complainers vs. 11 controls (age = 56.8 vs. 57.1)	MFG (BA10/9), ACC (BA24/32), insula (BA 13), precuneus (increased activation as WM demand increased)	Caudate	(1) No group differences in behavioral performance
Hu et al. (2017) [22]	Future-oriented decision making	20 memory clinic SCD vs. 24 controls (age: 68.3 vs. 66.49)		Medial frontal polar cortex, ACC, insula	(2) SCD showed reduced future-oriented choices (3) Positive correlation between hippocampal activation (ROI analysis) and future-oriented choice in only the control.
Hayes et al. (2017) [21]	Memory: successful vs. unsuccessful encoding	23 SCD vs. 41 controls (age: 68.6 vs 67.5) 21 out of 23 were memory clinic SCD	(1) Negative subsequent memory effect in the occipital lobe, SPL, PCC (2) More complaints, more negative subsequence memory effects in DMN (PCC, precuneus, VMPFC)		(1) No group differences in behavioral performance

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
