# Peer review of "A Homeostatic Model of Subjective Cognitive Decline"

_brainsci, 2018, doi:10.3390/brainsci8120228_

Reviewer 1 Report

This is a well-written manuscript which reviews subjective cognitive decline (SCD) and presents a new model of SCD that centers on prediction error monitoring. This model has potentially important implications for the study of SCD and approaches for intervention.  The manuscript may be improved by addressing the following comments:

Additional support for the focus on error prediction monitoring may be useful to help the reader understand the rationale for the model. The authors may be able to clarify how their model explains the relationship between the “sense of errors” discussed in the manuscript and the experience of cognitive tasks being more effortful in SCD. Often patient complain about working harder to maintain performance – this would fit with a homeostasis model, but is not clearly integrated with the prediction error model.  The abstract frames the proposed model as an extension of compensatory hyperactivation models; additional discussion of how these model interact would be useful in fitting the proposed model in the context of the current work in SCD.

In section “8. Homeostasis Breakdown”, the analogy of temperature regulation is helpful in the first paragraph but may distract readers throughout the remainder of the section from the homeostatic model being discussed.

The conclusions section could be strengthened by including a more in-depth discussion of future work. The authors should propose lines of research that would test the proposed model, including a discussion of method to better assess error prediction monitoring and homeostatic processes in the brain.  Using neuroimaging, how would they address these questions?  How would they test theories of monitoring vs control processes?

Author Response

1.    Additional support for the focus on error prediction monitoring may be useful to help the reader understand the rationale for the model. The authors may be able to clarify how their model explains the relationship between the “sense of errors” discussed in the manuscript and the experience of cognitive tasks being more effortful in SCD. Often patient complain about working harder to maintain performance – this would fit with a homeostasis model but is not clearly integrated with the prediction error model.  The abstract frames the proposed model as an extension of compensatory hyperactivation models; additional discussion of how these models interact would be useful in fitting the proposed model in the context of the current work in SCD.

a.    The increased effort necessary to maintain cognitive performance may be in itself a form of prediction error. For example, an individual may be used to finishing crossword puzzles in 20 minutes (prediction). However, they might find that they are now needing 30 minutes or more to complete a puzzle, reflecting both increased effort and prediction error. This has been clarified in Section 6of the revised manuscript.

b.    Discussion of the interaction between compensation and homeostatic models has been added to Section 8.

2.    In section “8. Homeostasis Breakdown”, the analogy of temperature regulation is helpful in the first paragraph but may distract readers throughout the remainder of the section from the homeostatic model being discussed.

a.    We have reduced the discussion of thermoregulation as suggested.

3.    The conclusions section could be strengthened by including a more in-depth discussion of future work. The authors should propose lines of research that would test the proposed model, including a discussion of method to better assess error prediction monitoring and homeostatic processes in the brain.  Using neuroimaging, how would they address these questions?  How would they test theories of monitoring vs control processes?

a.    As suggested, more discussion of future work has been added to Future directions (Section 9).

Reviewer 2 Report

The authors review evidence for the neuroanatomical and neuropathological bases of subjective cognitive decline (SCD).  This is an interesting topic, and they provide a relatively clear and concise synthesis of the literature for readers unfamiliar with it. 

Although I enjoyed reading the paper, I feel that ultimately it promised more than it delivered.  In particular, after surveying the literature to formulate a plausible neurobiological substrate for SCD, the authors hypothesize a homeostatic “model” to account for the compensatory activations they have previously described, and suggest it might also contribute to the neurodegenerative process.  However, they do not provide any evidence for this very speculative proposition, and it appears to come ‘out of the blue’.  It more of a metaphor than a model, but even their example of thermoregulation seems off the mark.  Thermoregulatory systems have an established thermoneutral range bounded by thresholds that trigger thermoeffectors when exceeded.  This system is understood in some detail, including its working elements.  To extend this metaphor to SCD, what is analogous to the thermoneutral zone, what is its neurobiological basis, what are the sensor(s) that activate the compensatory responses, and if the compensatory response is corrective does the system turn off (as it does in thermoregulation)?  I think it is too speculative to posit a self-regulatory process at this point, although it might be reasonable to discuss this as a possibility in the section of future directions.

How were the citations reviewed in this paper identified and selected?  The authors do not state that a literature search was conducted – what was the process for deciding which citations to include?

The paper is generally well-written and the concepts are concisely and clearly elaborated – not an insubstantial achievement given the complexity of the topic.  However, the paper is riddled with numerous and frequent grammatical errors; individually, each is very minor, but in the aggregate they create a slightly aversive reading experience.

Author Response

1.     Although I enjoyed reading the paper, I feel that ultimately it promised more than it delivered.  In particular, after surveying the literature to formulate a plausible neurobiological substrate for SCD, the authors hypothesize a homeostatic “model” to account for the compensatory activations they have previously described, and suggest it might also contribute to the neurodegenerative process.  However, they do not provide any evidence for this very speculative proposition, and it appears to come ‘out of the blue’.  It more of a metaphor than a model, but even their example of thermoregulation seems off the mark.  Thermoregulatory systems have an established thermoneutral range bounded by thresholds that trigger thermoeffectors when exceeded.  This system is understood in some detail, including its working elements.  To extend this metaphor to SCD, what is analogous to the thermoneutral zone, what is its neurobiological basis, what are the sensor(s) that activate the compensatory responses, and if the compensatory response is corrective does the system turn off (as it does in thermoregulation)?  I think it is too speculative to posit a self-regulatory process at this point, although it might be reasonable to discuss this as a possibility in the section of future directions. 

a.     Evidence from several large cohorts have provided support for homeostatic dysregulation in multiple systems (e.g. lipids, immune, oxygen transport, liver function, vitamins, and electrolytes)—See Li et al. 2015 (new reference 52). The authors suggest that homeostatic dysregulation is not limited to those systems, and as such, we propose here that homeostatic dysregulation may be also occurring in cognition and SCD. We agree that further research is necessary to identify the set point, neural control system(s), and the sensor(s) that detect prediction error. We have revised the manuscript to include these points in Section 8 and 9. 

2.     How were the citations reviewed in this paper identified and selected?  The authors do not state that a literature search was conducted – what was the process for deciding which citations to include?

a.     In this paper, we performed a narrative review of the literature instead of a systematic approach. Papers were selected based on their relevance to the structural and functional correlates of SCD. This has been clarified in the abstract and introduction of the revised manuscript. 

3.    The paper is generally well-written and the concepts are concisely and clearly elaborated – not an insubstantial achievement given the complexity of the topic.  However, the paper is riddled with numerous and frequent grammatical errors; individually, each is very minor, but in the aggregate, they create a slightly aversive reading experience.

a.    The manuscript has been further proofread to improve readability of the paper.

Reviewer 3 Report

This manuscript describes the process and several theories to explain subjective cognitive decline, which is an early symptom of the onset of dementia, mild cognitive impairment and eventual Alzheimer's Disease.

The manuscript has a number of grammatical errors and textual inconsistencies which make reading the paper difficult, and in some place simply impossible to interpret without rewriting the authors' grammar. a partial list includes:

line 78: one which observed

line 156: regardless of the

line 164: in a varying level

line 165: simply cannot decipher it

line 182: is a general

line 184: inter to the

Lines 189-190: too messed up to correct

line 213: their an individual's

line 223: is a theory

line 237: comment incorrect based on examples given: should be vasoconstriction not shivering

line 250; overwork the neural system <--this would be highly controversial to most neuroscientists

Author Response

1.    The manuscript has a number of grammatical errors and textual inconsistencies which make reading the paper difficult, and in some place simply impossible to interpret without rewriting the authors' grammar.

a.    The manuscript has been proofread, and the listed grammatical errors have been corrected.

Reviewer 4 Report

Review of “A Homeostasis Breakdown Model of Subjective Cognitive Decline”

 This manuscript provides an interesting and useful overview of subjective cognitive decline (SCD) and the relation to prediction error. I have the following suggestions for the revision of this manuscript:

1.     On the one side, it seems that SCD can be a useful predictor for subsequent cognitive decline and dementia. This suggests that people are well calibrated on a metacognitive level and can reliably monitor prediction errors. On the other side, we would expect that in particular people with cognitive problems will have a hard time to accurately monitor their performance and environment. How can these seemingly contradictory findings be reconciled? This could be discussed in the text. Moreover, it would be useful to add a brief section in which healthy older adults’ metacognitive calibration is discussed. That is, to what extent do *healthy* older adults (that will not develop dementia in the near future) complain about memory problems and how good is their calibration of errors in the first place?

2.     The sample sizes -- and consequently the statistical power -- of the reviewed studies were small; this should be critically mentioned somewhere in the text: Inconsistencies between studies (e.g., concerning frontal hyper- vs. hypoactivation) could partially be due to sampling error.

3.     In section 8, “homeostasis breakdown” not only the role of hyperactivation but also of hypoactivation should be dialectically discussed: the authors state that “hyperactivation is the main homeostatic process that we are currently aware of in cognition” but there are also studies reporting hypoactivation (as mentioned in Table 1).

Other points:

 4.     The manuscript is too heavy on the use of abbreviations/acronyms. The use of acronyms should be reduced because they negatively affect the readability of any text. Rule of thumb: Use an abbreviation at least three times in a paper if you are going to use it at all. Otherwise spell out the term every time.

5.     The manuscript must be proofread again. Words are missing or misplaced. Examples:

5.1.  “Unlike other studies, their study is the only one observed group differences in the task performance.”

5.2.  Abstract. “A cognitive prediction error is a discrepancy between an implicit cognitive predictions and the corresponding outcome.”

6.     Table 1 is not well prepared and the entries in Table 1 are not well aligned which makes it difficult to read the table. This must be fixed in a revision.

7.     P.3 “There are two fMRI studies with non-memory tasks. Dumas and colleagues [27] investigated the executive functioning in SCD by using the n-back working memory task.” This should be rephrased. The authors probably mean “non-episodic memory” tasks?

8.     Line 114: “loss of integration of memory system”. This aspect has not been introduced in the text before and it is unclear at this position what it means. Please clarify.

9.     I felt that section 7 “Theories of the neural basis of SCD” was slightly misplaced in the manuscript and should be presented earlier (e.g., along with sections 2/3)

Author Response

1.    On the one side, it seems that SCD can be a useful predictor for subsequent cognitive decline and dementia. This suggests that people are well calibrated on a metacognitive level and can reliably monitor prediction errors. On the other side, we would expect that in particular people with cognitive problems will have a hard time to accurately monitor their performance and environment. How can these seemingly contradictory findings be reconciled? This could be discussed in the text. Moreover, it would be useful to add a brief section in which healthy older (that will not develop dementia in the near future) complain about memory problems and how good is their calibration of errors in the first place?

a.    These findings may not necessarily be contradictory. For example, cognitive monitoring may require a certain level of cognitive functioning. After the onset of objective cognitive decline, cognitive monitoring abilities may decrease as well, following an inverted U-shaped pattern). This has been added to Section 6 of the revised manuscript.

b.    By definition, individuals with SCD notice a change in cognitive performance, despite normal cognitive testing. The typical situation is an older adult who performs normally on objective memory tests, yet reports that their memory has declined. It is possible the subjective reporting is biased by errors in memory, i.e., they are misremembering how good their memory used to be. However, this seems an unlikely scenario as on objective memory tasks their memory is unimpaired. It is possible that the discrepancy they are reporting is driven by anxiety, neuroticism, or other psychological factors, and thus not necessarily reflective of a change in memory performance. The homeostatic prediction model can give a framework for understanding how this subjective perception of performance relates to objective performance.

2.    The sample sizes -- and consequently the statistical power -- of the reviewed studies were small; this should be critically mentioned somewhere in the text: Inconsistencies between studies (e.g., concerning frontal hyper- vs. hypoactivation) could partially be due to sampling error.

a.    Statistical power and sampling errors are critical limitations of prior work in SCD. We have thus added a statement to reflect this.

3.    In section 8, “homeostasis breakdown” not only the role of hyperactivation but also of hypoactivation should be dialectically discussed: the authors state that “hyperactivation is the main homeostatic process that we are currently aware of in cognition” but there are also studies reporting hypoactivation (as mentioned in Table 1).

a.    The studies reporting hypoactivation in Table 1 are not necessarily reporting compensatory hypoactivation, but rather that hypoactivation is associated with compensatory hyperactivation elsewhere. A caption has been to the table to clarify this.

4.    The manuscript is too heavy on the use of abbreviations/acronyms. The use of acronyms should be reduced because they negatively affect the readability of any text. Rule of thumb: Use an abbreviation at least three times in a paper if you are going to use it at all. Otherwise spell out the term every time. 

a.    We reduced the amount of abbreviations and acronyms as suggested.

5.    The manuscript must be proofread again. Words are missing or misplaced.

a.    Suggested revisions were made, and the manuscript was further proofread.

6.    Table 1 is not well prepared and the entries in Table 1 are not well aligned which makes it difficult to read the table. This must be fixed in a revision. 

a.    The table has been recreated and realigned.

7.    P.3 “There are two fMRI studies with non-memory tasks. Dumas and colleagues [27] investigated the executive functioning in SCD by using the n-back working memory task.” This should be rephrased. The authors probably mean “non-episodic memory” tasks?

a.    This has been corrected to “non-episodic memory” tasks.

8.    Line 114: “loss of integration of memory system”. This aspect has not been introduced in the text before and it is unclear at this position what it means. Please clarify.

a.    Loss of integration of the memory system can be unclear, thus we rephrased the results of that study without this phrasing.

9.    I felt that section 7 “Theories of the neural basis of SCD” was slightly misplaced in the manuscript and should be presented earlier (e.g., along with sections 2/3)

a.    We agree that this section should be presented earlier—it is now section 4 of the manuscript.

Round  2

Reviewer 1 Report

The authors have addressed our concerns and comments.